



# Observations of Tall-Building Wakes Using a Scanning Doppler Lidar

Natalie E. Theeuwes[1,2], Janet F. Barlow[1], Antti Mannisenaho[3], Denise Hertwig[1], Ewan O'Connor[3], and Alan Robins[4]

[1]Department of Meteorology, University of Reading, UK
[2]now at Royal Netherlands Meteorological Institute (KNMI), de Bilt, The Netherlands
[3]Finnish Meteorological Institute, Finland
[4]Department of Mechanical Engineering Sciences, University of Surrey, UK

**Correspondence:** Natalie E. Theeuwes (natalie.theeuwes@knmi.nl) and Janet F. Barlow (j.f.barlow@reading.ac.uk)

**Abstract.** High-rise buildings, increasingly a feature of many large cities, impact local atmospheric flow conditions. Tall building wakes affect air quality downstream due to turbulent mixing and require parametrization in dispersion models. Previous studies using numerical or physical modelling have been idealised and under neutral conditions. There has been a lack of data available in real urban environments due to the difficulty in deploying traditional wind sensors. Doppler wind lidars (DWLs) have been used frequently for studying wind turbine wakes but never building wakes. This study is a year-long deployment of a DWL in a complex urban environment studying tall building wakes under atmospheric conditions. A HALO Photonic Streamline DWL was deployed in a low- and mid-rise densely packed area in central London. From its roof-top position (33.5 m agl compared to mean building height 12.5 m), Velocity Azimuth Display (VAD) scans at zero-degree elevation intersected with two, taller nearby buildings of 90 and 40 m agl. Using an ensemble averaging approach, wake dimensions were investigated in terms of wind direction, stability and wind speed. Boundary-layer stability categories were defined using eddy covariance observations from the BT Tower (191 m) and mixing height estimations from vertical stare scans. A method for calculating normalised velocity deficit from VAD scans is presented. For neutral conditions, wake dimensions around both buildings for the prevailing wind direction were compared with the ADMS-Build wake model for a single, isolated cube. The model underpredicts wakes dimensions, confirming previous wind tunnel findings for the same area. Under varying stability, unstable and deep boundary layers were shown to produce shorter, narrower wakes. Typical observed wake lengths were 120-300 m and widths were 80-150 m and were reduced by 50-100 m downwind. Stable and shallow boundary layers were less frequent and produced an insignificant difference in wake dimensions to neutral conditions. The sensitivity to stability was weakened by enhanced turbulence upstream (i.e., due to other building wakes). Weakened stability dependence was confirmed if there were more obstacles upstream as the wind direction incident on the buildings changed. The results highlight the potential for future wake studies using multiple DWLs deploying both vertical and horizontal scan patterns. Dispersion models should incorporate the effect of a complex urban canopy within which tall buildings are embedded.



## 1 Introduction

As the urban population increases, cities grow both horizontally and vertically. High-rise buildings, increasingly a feature of
many large cities, impact local atmospheric conditions. Tall building wakes determine near-surface wind climate and pollutant
dispersion as they alter momentum and scalar exchange within and above the urban canopy. However, they have mostly been
studied under idealised conditions using numerical modelling (e.g. Nozu et al., 2015; Liu and Niu, 2016) or physical modelling
(e.g. Sheng et al., 2018; Mishra et al., 2023). Other studies have focused on isolated tall buildings (Nozu et al., 2015), or
examined flow interference effects between a few tall buildings for wind loading considerations (Lam et al., 2008). Hence,
there is a lack of research into tall building wakes embedded in a real urban canopy under varying atmospheric stabilities.

The impact of tall buildings on local flow and pollutant dispersion within the surrounding urban canopy is significant (Heist
et al., 2009; Brixey et al., 2009) and extends several streets away (Fuka et al., 2018; Aristodemou et al., 2020). The urban
canopy also alters the wake structure: Hertwig et al. (2019) used wind tunnel experiments under neutral conditions to study
the influence of the surrounding urban roughness on tall building wakes. Compared to an isolated tall building they found
the turbulent structure of wakes from high-rise buildings within urban canopies to be altered and wake dimensions to be
dependent on upstream urban roughness. Huang et al. (2021) used CFD simulation to show that street-level pollutant dispersion
is enhanced compared to the case with no tall buildings. They found that the wake also alters significantly in size and turbulence
intensity when atmospheric stability is varied.

Confirming these results by measurements in real urban areas is challenging due to restrictions in instrument placement and
representativeness within a complex microclimate (Oke, 1998; Stewart, 2011). Measuring tall building wakes using mast-based
instrumentation is logistically unfeasible. However, remote sensing methods are well-suited to profiling urban boundary layers
(Barlow et al., 2011a; Kotthaus et al., 2022). Doppler lidars have been used to measure boundary layer depth and turbulence
profiles (Barlow et al., 2015; Kongara et al., 2012) under different atmospheric stabilities and analyse the response of the
urban wind profile to surface roughness (Drew et al., 2013; Kikumoto et al., 2017; Ortiz-Amezcua et al., 2022; Filioglou et al.,
2022). Dual Doppler lidars have potential for improving pollution dispersion models (Collier et al., 2005) due to more accurate
retrieval of the wind field in highly turbulent urban flows. Calhoun et al. (2006) used dual lidars to create "virtual towers"
upstream of a cluster of tall buildings and observed deceleration of the vertical wind profile due to their drag.

Doppler lidars have increasingly been used to observe wind turbine wakes to validate instantaneous wake models for wind
loading assessment and wind-farm layout optimisation (see review by Sun et al., 2020). They can be nacelle-mounted (Bingöl
et al., 2010; Trujillo et al., 2011) or ground-based, using combinations of profiling and scans to build up a 3D picture of the
wake (Banta et al., 2013; Barthelmie et al., 2014, 2018). Dual Doppler lidars have been used to give greater accuracy of wind
retrieval in the highly turbulent wake (Iungo et al., 2013; Iungo and Porté-Agel, 2014) or non-homogeneous flow over complex
terrain (Vasiljević et al., 2017). Single scanning lidars have also been effective in analysing wake dimensions (Aitken et al.,
2014; Bodini et al., 2017).

Wind turbine power output has a complex dependency on stability (Wharton and Lundquist, 2012) due to wind shear and
turbulence, however, overall wind farm performance is reduced in stable conditions (Hansen et al., 2011) as wakes are longer





and more intense due to a lack of turbulent mixing to erode the velocity deficit. However, Aitken et al. (2014) found little dependence of wake structure on atmospheric stability or upstream turbulence intensity, suggested to be due to the heterogeneous upstream terrain reducing the range of stability. Wildmann et al. (2020) found that observed wake turbulence intensity was

higher than upstream turbulence intensity until the latter was around 25%. It remains to be seen whether these results relate to tall building wakes in warmer, rougher urban areas where the stability range is smaller.

To the authors' knowledge, this is the first study to analyse a long time series of Doppler lidar observations of tall building wakes at an urban site under different atmospheric conditions. The study took place in central London as part of the MAGIC field campaign (Song et al., 2018), of which the measurements used here are described in Section 2. In Section 3 the wind,

stability and mixing height climatology for the site is presented. In Section 4 a new methodology for defining velocity deficits using a spatially-averaged windfield is described. Wakes during neutral conditions are described in Section 5 and in Section 6 the impacts of stability, wind speed and wind direction are presented.

## 2 Measurements

### 2.1 Site Description

Measurements took place in the London Borough of Southwark, just south of the River Thames, as part of fieldwork associated with the MAGIC project (Song et al., 2018, see Fig. 1). Instruments were located on top of a building belonging to London South Bank University (LSBU, 51°29'53.4"N; 0°06'07.2"W) of where the roof height is 36.5 m above sea level, asl (33.5 m above ground level, agl). The study area consisted of low-rise residential and mid-rise office and commercial buildings with occasional tall buildings. The mean canopy height $z_h$= 12.8 m, standard deviation $\sigma_h$ = 9.5 m, and plan area index $\lambda_p$=0.54.

Therefore, the lidar was located at about $2.6z_h$ and most likely within the roughness sub-layer.

In the direct vicinity of the lidar there are several tall buildings, especially to the southeast. Between bearings of 080 to 135° a plant room on the roof of the building prevents the lidar from retrieving reliable signal when scanning horizontally (see Fig 1(b), left hand side). Between 135 and 175° a cluster of taller buildings between 100 and 400 m away also limit the range of retrievable signal. Therefore, this paper will focus on tall buildings to the north-west of the lidar: in particular, two buildings

approximately 200 m and 300 m away from the lidar, building A and B respectively. Building A has a height of 93 m a.s.l. (90 m a.g.l.). Therefore, the measurement height of the horizontal scans agl (Sec. 2.2) is $0.37 \cdot H_A$. Building B has a height of 43 m a.s.l. (40.2 m a.g.l.), and the measurement height is $0.83 \cdot H_B$. Hertwig et al. (2019) studied the wake of Building A for northerly flow simulated in a wind tunnel.

### 2.2 Doppler Lidar

A 1.5$\mu$m pulsed, heterodyne Doppler lidar (HALO Photonics Streamline) was located on the roof-top (pulse repetition rate 10 kHz, pulse duration $2.0 \cdot 10^{-7}$ s). The lens diameter is 75 mm and the focal length was set to infinity for all scan patterns. The lidar was configured with a range resolution of 18 m with 6 points per range bin. The sampling frequency is 50 MHz, the



velocity resolution was 0.382 m/s, and the Nyquist velocity was 19.456 m/s. Measurements were valid from a range of 54m, returns from the first three gates being spurious due to the geometry of transmitter and receiver. The lidar was aligned with the long axis of the building (bearing 140°) to ensure a precise orientation. All data have been rotated towards true north for the analysis.

Data from 7 March 2019 to 31 March 2020 are used for the analysis in this study. During this year, the lidar was almost fully operational with the exception of 4 days in September 2019. A long period of data was required over a large range of atmospheric stability, mixing height, wind speed and direction.

Before 9 September 2019 the integration time per ray was 1 s for stare mode, 2 s for VAD scans and maximum range was 200 gates. After that date, integration time was changed to 3 s to improve the signal to noise (SNR). Maximum range was changed to 555 gates to allow correction of instrument background shape according to Manninen et al. (2016). Analysis here is confined to the lowest part of the boundary layer where observations were rarely limited by SNR due to high pollution levels, therefore the accuracy of the wind retrievals before and after the change was comparable.

Three scanning modes were used: a 6-point Velocity Azimuth Display (VAD) scan at 75° elevation (every 6 minutes), a 72-point VAD scan at 0° (every 12 minutes), and the remainder of the time in vertical stare mode. The VAD scan at 0° elevation intersects with obstacles higher than 36.5 m a.s.l. (as shown in Fig 1(a)).

The Halo Lidar Toolbox, developed at the Finnish Meteorological Institute, was used to process retrieved data (Manninen, 2019). A description of all modules in the processing chain is given in Ortiz-Amezcua et al. (2022), only ones applied in the current study are described here:

1) Background corrections were applied to raw signal intensity according to Manninen et al. (2016) and Vakkari et al. (2015). Instrumental precision of radial velocities was estimated with the method given by Pearson et al. (2009); Rye and Hardesty (1993). Attenuated backscatter coefficient was computed with uncertainty according to Manninen et al. (2018).

2) Wind vector profiles were calculated from VAD scans by assuming a stationary and horizontally homogeneous wind field and using a least squares method (Päschke et al., 2015) with uncertainties estimated using the method of Newsom et al. (2017).

Mixing height was derived from vertical velocity measurements taken in vertical stare mode. Vertical velocity variance was used to calculate the mixing height as in previous studies (Barlow et al., 2015; Halios and Barlow, 2018; Theeuwes et al., 2019a). Statistics were calculated in periods of 30 minutes. Data where $SNR + 1 < 1.01 (< 20dB)$ were filtered out and only 30 minute periods with more than 30% of datapoints available were accepted. Mixing height was determined as the height in the vertical profile where the velocity variance dropped below a threshold. Following Barlow et al. (2015), 21 thresholds around 0.1 $m^2s^{-2}$ (0.069 to 0.129 $m^2s^{-2}$) were used to evaluate uncertainty due to choice of threshold. The final mixing height was taken to be the median of the 21 values.

## 2.3 Flux Measurements

Eddy covariance measurements at two sites were used in this study. The first is located on top of the British Telecom (BT) Tower (191 m agl, 218 m asl, 51°31'17"W, 0°08'20"N) which is 3.6 km to the northeast of the LSBU site. Flux data presented here were measured by a sonic anemometer (R3-50, Gill Instruments Ltd) with a sampling rate of 20 Hz. For a full description



of other instruments and the site see Lane et al. (2013). The instruments were placed on a mast on top of an open-lattice scaffolding tower of height 12.3 m to minimise flow distortion (Barlow et al., 2011b). Turbulent fluxes were calculated in half-hourly averages using the methodology set out by Wood et al. (2010).

The second set of instruments was located near to the lidar on the LSBU roof-top (see Fig 1 b). A 3D sonic anemometer (R3-50, Gill Instruments Ltd) was logged at 20 Hz and mounted on a mast so that the centre of the measurement head was 3.83 m above the roof. Alongside this was an automatic weather station, logged at 1Hz (Vaisala WXT520) whose uppermost 2D sonic anemometer and rain sensor were at 2.96 m. Due to limited options for safe installation of the mast it was located to the west of a plant room of approximate height 1.6 m. It is acknowledged that the LSBU turbulence measurements are well within

the roughness sub-layer and influenced by the plant room for easterly winds. Placing a flux tower within the inertial sub-layer (ISL) is unfeasible practically as the minimum height of the ISL is likely to be at least twice maximum building height, i.e., around 180 m based on Building A.

All data are averaged in 30-min periods, including the lidar wind profiles derived from the VAD 75° scans every 6 minutes and the 2–3 VAD 0° scans measured every 12 minutes. Processing all data in 30-minute averages facilitates straightforward

filtering of periods based on mixing height, stability at the BT Tower site, or other meteorological parameters as described in the following sections.

## 3    Wind and Stability Climatology

The wind rose above the roughness sub-layer at around $2.5 \cdot H_A$ (Fig. 2(b)) was derived from the VAD 75° scans by averaging wind speed from three 18 m gates, i.e. from 195.5 m to 249.5 m agl with a midpoint height of 222.5 m agl. During the

measurement year, the predominant wind direction at the site was west – southwesterly, with a median windspeed of 7.7 m s$^{-1}$. The 0° elevation scan intersects with taller buildings, suggesting that the height of this scan (36.5 m asl) is in the roughness sub-layer. Hence, a spatially-averaged wind vector was approximated by a) fitting a sinusoid to radial velocities with azimuth angle to give a wind vector at each range gate (Päschke et al., 2015), and b) averaging the wind vectors across all range gates out to 500 m. An example of this method is shown in Fig. 4. Fig. 2(a) shows that the wind speed at 36.5 m is lower compared to

222.5m (median windspeed = 2.6 m s$^{-1}$) and the most frequent wind direction is backed slightly (south-westerly). Compared to the wind-rose at 222.5 m, there is an increase in the frequency of southwesterly and northeasterly winds and a decrease in northwesterly and southeasterly winds. As Fig. 1 shows, most tall buildings are located to the northwest and southeast of the site, which might create a neighbourhood-scale channelling effect.

The effect of atmospheric conditions on building wakes will be analysed according to two measures. The urban Monin

Obukhov stability parameter was calculated using BT Tower eddy covariance measurements, where $z - z_d = 180$ m (displacement height $z_d$ estimated to be 11 m) and $L$ is the Obukhov length. The frequency of different stability classes was as follows: very unstable ($\frac{z-z_d}{L} < -0.5$, 27.4 %), unstable ($-0.5 < \frac{z-z_d}{L} < -0.1$, 17.5 %), near neutral ($-0.1 < \frac{z-z_d}{L} < 0.1$, 17.1 %), stable ($0.1 < \frac{z-z_d}{L} < 0.5$, 15.9 %), and very stable ($\frac{z-z_d}{L} > 0.5$, 21.6 %). Most stability classes occur in similar frequencies, although very unstable cases occur most often.





When conditions are stable according to the local measurements at the BT Tower, they may be located in the residual layer above a shallow, night-time boundary layer which can even be weakly convective due to delayed heat release from the urban surface (Barlow et al., 2015; Halios and Barlow, 2018). It should be noted that temperature profiling by radiosondes, tethersondes or RASS is not permitted in central London. Therefore, the mixing height is also used to categorise atmospheric conditions. An estimate of the mixing height $z_{\mathrm{MH}}$ was derived from the vertical velocity variance as explained in Sect. 2.2. As flow patterns around buildings can be sensitive to the height of the building relative to the boundary layer depth (Bächlin et al., 1983) the height of building A ($H$) is divided by the mixing height ($z_{\mathrm{MH}}$). Here, three classes are chosen: shallow mixing height that is no more than twice the building height ($\frac{H}{z_{\mathrm{MH}}} > 0.5$, 14.7 %), moderate mixing height ($0.1 < \frac{H}{z_{\mathrm{MH}}} < 0.5$, 31 %) and deep mixing height ($\frac{H}{z_{\mathrm{MH}}} < 0.1$, 13 %)

The method of retrieving the mixing height cannot return an accurate result in cases where the vertical velocity variance threshold is not reached. This can occur when the variance is less than 0.1 m$^2$s$^{-2}$ in the lowest gate where signal can be retrieved (gate 4, mid-point height 98.5 m agl) which is often the case when there is a very stable, shallow boundary layer. Alternatively, mixing height is not determined when the SNR reduces below the threshold for ensuring data quality but the vertical velocity variance has not yet reduced below 0.1 m$^2$s$^{-2}$. This may occur when clean air is entrained at the top of a deep convective boundary layer, causing low SNR, or when there is a cloud present, causing large attenuation of signal. Filtering out these cases (41.2 %) reduces uncertainty in the ratio $\frac{H}{z_{MH}}$ and analysis in the following sections is done on the remaining 58.8 % of 30 minute time periods.

As mixing height is strongly correlated with stability, the two measures are combined to classify the boundary layer affecting wakes: stable and shallow, neutral and moderate, and unstable and deep (Fig. 3). Fig. 3 shows that for the data filtered by available mixing height, the predominant wind direction is still southwesterly, with approximately equal partitioning between the three boundary layer classes. For westerly to northerly directions there are few cases where the boundary layer is stable and shallow. This is in contrast to easterly wind directions where the majority of the cases have a stable and shallow boundary layer. In the United Kingdom, easterly flow is often associated with high pressure, stable conditions and shallow boundary layers due to enhanced subsidence. During the first part of this study the focus lies on the predominant wind direction ($\phi \approx 220°$) as there is a large amount of data and all stability classes are observed in approximately equal frequencies.

## 4 Analysis of Horizontal VAD Scans at 0° Elevation

In order to identify building wakes in the VAD 0° scans, the velocity deficit is estimated from the radial velocity field. Velocity deficit can be defined as the decrease in velocity behind an object compared to the undisturbed flow (Hertwig et al., 2019) or upstream/ambient flow (Aitken et al., 2014), which are not available in the present study. Instead, a neighbourhood scale spatially-averaged velocity is estimated and the derivation is illustrated in Fig. 4. The instantaneous radial velocity as a function of range and azimuth angle $V_\theta(r,\theta)$ for one scan is shown in Fig. 4a. Negative and positive radial velocities are towards and away from the lidar respectively, hence the flow is northeasterly in the example. Next, the wind vector was derived for each range gate using a least squares method (Paeshke) to fit a sine function to the rays of the scan (every 5° in azimuth) for



which there was a good retrieval. The wind vectors were then averaged across range out to a chosen radius (typically 500 m) around the lidar to give a neighbourhood scale spatially-averaged velocity, $\overline{V}$, from which spatially-averaged radial velocity

was calculated using

$$\overline{V_\theta} = \overline{V}\sin(\theta - \phi + 3\pi)V. \tag{1}$$

where $\theta$ is the azimuth angle and $\phi$ the wind direction. Projecting $\overline{V_\theta}$ across all gates gives the radial velocity field pertaining to the spatially-averaged velocity, i.e. what the lidar would measure if the wind field were homogeneous (Fig. 4c). The instantaneous velocity deficit field is then defined as $V_\theta - \overline{V_\theta}$ (i.e. Fig. 4a minus Fig. 4c) which is then normalised by the spatially

averaged radial velocity $\overline{V_\theta}$ (Fig. 4d).

Buildings that intersect the lidar beam are clearly visible in the attenuated backscatter coefficient (Fig. 4(b)) where they are denoted by $\beta > 10^{-4} m^{-1} sr^{-1}$. Obstacles are thus simply located by applying this threshold. Obstacles can be buildings, trees or masts and may vary throughout the measurement period, as trees lose their leaves and new structures are built throughout the measurement year.

Southeast of the lidar there is no signal, because a plant room and other instruments on the roof blocked the field of view of the lidar (see Fig. 1(b)). Therefore, this study will focus on building wakes to the north and west of the lidar. Behind obstacles no reliable signal was returned and these radial velocities are filtered out. The velocity deficit of a single scan varies greatly spatially (Fig. 4d) and wakes of individual obstacles are not easily distinguishable. Given that building wakes persist over time whereas passing turbulent coherent structures are transient, averaging over an ensemble of scans is proposed.

When averaging an ensemble of radial velocity scans for the predominant wind direction where $\phi = 220° \pm 10°$, the effect obstacles have on the flow becomes clearer (Fig. 5). Most of the obstacles identified by high attenuated backscatter values show a wake in the radial velocity field along the predominant wind direction. As mentioned in Sec. 2.1, the focus will be on buildings A and B as these are closest to the lidar where there is better data quality and resolution, and generally the wakes were not obstructed by other objects. Fig. 5(b) shows the ensemble mean of the normalised velocity deficit. Both buildings

clearly exhibit a wake oriented stream-wise with the wind direction. To the south of building A there is a speed-up of the wind. For this wind direction the wake of building B possibly affects the wake of building A due to deceleration upstream of building A.

By comparison with Fig. 1(a), wakes from other buildings can be identified. The 75 m tall building approximately 580 m to the west of the lidar also shows a distinct wake stretching at least 300 m downstream. To the south of this building there are

two smaller obstacles that also generate a disturbance in the flow. At 500 m southwest of the lidar is a small cupola on top of the Imperial War Museum. The weaker downstream influence of these smaller obstacles can nevertheless be detected in the ensemble mean.

Given that building wakes can be identified in the ensemble-averaged VAD $0°$ scans, we now choose to quantify wake dimensions using ensemble means rather than instantaneous scans (as used in wind turbine studies to identify wind patterns

associated with loading). This is more relevant to pollution dispersion over longer timescales.



## 5 Wake Characteristics During Neutral Conditions

The magnitude of the velocity deficit during neutral conditions will now be assessed. Since Hertwig et al. (2019) do not report wind tunnel measurements for the predominant wind direction which would be directly comparable, we follow their analysis by comparing wake data with the ADMS-Build model, which is a small-deficit wake model based on constant eddy-viscosity

theory (Robins and McHugh, 2001). For a full description of the model, see Appendix 1 of Hertwig et al. (2019). The building dimensions used in the calculation are height (A = 93m, B = 41m), width (A = 28m, B = 42m) and length (A = 33m, B = 42m), taken from the data used for Fig. 1(a).

As inputs, the ADMS-Build model requires the wind speed at building height H, which is here approximated by the spatially-averaged wind speed $\overline{V_\theta}$ from each instantaneous VAD 0° scan. Also, the 30 minute averaged friction velocity at the LSBU

roof-top site. The friction velocity at the LSBU site is a measurement in the roughness sublayer and may be underestimated compared the friction velocity in the inertial sublayer (Rotach, 1999; Kastner-Klein and Rotach, 2004). However, the local flow around the buildings could enhance the friction velocity (e.g. Christen et al., 2009; Theeuwes et al., 2019b). Using the friction velocity from the BT Tower site at 191m height had a negligible difference in the magnitude of the modelled wake (not shown).

Fig. 6 shows that the ADMS model underestimates the intensity and physical extent of the normalised velocity deficit due to the wakes of Buildings A and B, at relative heights $0.37 \cdot H_A$ and $0.83 \cdot H_B$. The intensity underestimate might partly be explained by the wind-speed input being less than the wind speed at the building, however, the modelled lateral spread is narrower and decay downstream is more rapid than the observed wake. Hertwig et al. (2019) also showed an underestimation of the velocity deficit intensity and spread by ADMS compared to wind tunnel measurements, especially close to the urban

canopy. Their measured wind profiles showed that the wake structure is altered by the presence of the roughness sublayer, which is not represented in the simple ADMS wake model. A full test of the model formulation requires three-dimensional measurements to explore whether diffusivity is enhanced due to urban canopy turbulence, causing wakes to spread further. Whilst the present result is limited, it confirms the trend for model underestimation from the (Hertwig et al., 2019) study. Nevertheless, it is encouraging that a simple wake model can capture such complex wakes to some degree.

## 6 Sensitivity of Wake Characteristics To Atmospheric Conditions

Having presented a methodology for observing building wakes using Doppler lidar and shown that they exhibit similar dimensions to those modelled for a simple, cubic building under neutral conditions, we return to our hypotheses that stability and upstream roughness affect wake dimensions. The sensitivity to wind speed and direction are also tested in this section.

### 6.1 Effect of Stability on Wake Dimensions

We expect that enhanced upstream turbulence intensity leads to higher momentum diffusivity and therefore a velocity deficit that decays more rapidly with distance, i.e. a shorter wake (Castro and Robins, 1977). This might occur with larger upstream





obstacles, and/or more unstable atmospheric conditions. A second hypothesis is that the effect of stability would be less if upstream roughness is large (Aitken et al., 2014). As mentioned in Sec. 3, two parameters (mixing height and Obukhov length) were used to classify the boundary layer as stable and shallow; neutral and moderate; or unstable and deep. The differences

between wake structure for the three stability classes appear minor (Fig. 7 and 8). However, the wake of building B is shorter for deep and unstable boundary layers compared to the other two stability classes (Fig. 8d), with the normalised velocity deficit reaching zero by 100 m downstream. This is not the case for building A (Fig. 8b). From our prior hypotheses, it seems that the wake from building B enhances turbulence intensity upstream of building A, causing stability to impact wake dimensions less for building A. The wake of building A is also asymmetrical and is not directly parallel to the wind direction, exhibiting

accelerated flow just to the south–southeast. The magnitude of this flow acceleration is similar across all three stability classes.

## 6.2 Wind Speed

We also analysed the results by wind speed, using the spatially-averaged velocity derived from the 0° degrees elevation scans, $\overline{V}$. This analysis includes more data than the previous figure, because the data is independent of the mixing height, which is not always able to be classified. The expectation is that wake behaviour in light winds would be very variable as the flow

is unstable or stable, but behaviour should converge as wind speeds increase, turbulence intensity decreases, and flow tends towards neutral. However, the influence of the wind speed on the length, width, and magnitude of the wakes is not significant (Fig. 9). For building B, this is perhaps because the previously demonstrated dependence on unstable conditions is averaged out by stable conditions within the lower wind speed classes. These results are again similar to the study by Iungo and Porté-Agel (2014), who did not find large differences in the length of the wind turbine wake for different background wind speeds. This

provides confidence that the wake length of building B for $\phi \approx 220°$ is shorter due to unstable conditions.

## 6.3 Wind Direction

As the wind changes direction, the lidar can have a better viewing angle of the wake and better spatial resolution. This section addresses whether wake characteristics change with flow direction. Fig. 10 shows ensemble mean normalised velocity deficit for non-prevailing wind directions with sufficient scans, averaged over all stabilities. Fig. 11 and 12 show selected cross-

sections and their dependence on stability and wind speed. The wind direction used for filtering the data is the spatially averaged wind direction from the 0° elevation scans.

During a northerly flow the wake from building A is weak (Fig. 10a) due to the buildings upstream obstructing the flow. The building is almost symmetrical, therefore its shape is unlikely to explain a difference with oncoming flow direction. Figure 1(a) shows several tall buildings directly north of building A at distances of 500 and 600 m. The wake from building B is not

visible, as the wind direction is northerly and the radial velocity around 270° is near zero.

For a wind direction of 60° there is a much longer wake behind building A (Fig. 10b) compared to $\phi = 220°$ where building B wake was upstream. The streamwise cross-section shows that the wake takes up to approx. 300 m to decay towards the spatial mean velocity (Fig. 11b). There are very few tall obstacles obstructing the flow upwind of building A. The results suggest a



longer, wider, stronger wake with higher wind speeds (Fig. 12 a and b), and that unstable conditions produce a shorter wake (Fig. 11b), which supports the finding for $\phi = 220°$.

For a wind direction of 250° the wake from building B is shorter and weaker than for 220° (Fig. 10b compared to Fig. 8), shows a similar sensitivity to stability (Fig. 11b) and strengthens with wind speed (Fig. 12b). The width of the building presented to the wind is similar, to 220° therefore the width is similar.

For a wind direction of 290°, which is uncommon, there are fewer scans but similar dependencies are found for the wake of building A (Figs. 10e, 11e-f, and 12e-f). The data that is available shows unstable and low wind speed conditions lead to a shorter and narrower wake compared to neutral and high wind speed conditions.

Finally, for a wind direction of 315° (Figs. 10f, 11g-h, and 12g-h) building A is exactly upwind of the lidar, leading to maximum changes in radial velocity and a well-resolved wake. There is a less clear dependency on stability or wind speed. However, the wake is considerably shorter than for $\phi = 290°$: there is a tall building c. 600 m directly upstream of A that may enhance turbulence intensity (Fig. 1(a)). The regions of speed-up on each side of building A indicated by positive values of velocity deficit are stronger and the wake is narrower, perhaps influenced by channelling flow induced by building B.

Overall, analysis of the data categorised by wind speed is affected by uneven and small samples across the categories. Nevertheless, the results broadly support the conclusions summarised in the previous section, that unstable conditions produce shorter wakes, neutral conditions with higher winds speeds produce longer wakes. However, the most notable conclusion might be that inhomogeneity in the built environment is often the dominant factor and changes with wind direction. Boundary layer stability and wind speed only appear to influence wake behaviour when the fetch upwind of the obstacle is relatively homogeneous.

## 7    Conclusions

Observations of tall building wakes in urban areas are important for determining their impact on local wind microclimate and pollutant dispersion. To our knowledge, this is the first year-long study to observe tall building wakes in a real and complex urban environment under atmospheric conditions using a Doppler lidar. The results from this study can be used to inform dispersion modelling and numerical weather prediction.

A Doppler lidar was located in central London at a height of 33.5 m above ground level, while the dense low to mid-rise urban canopy surrounding the site had a mean height of 12.8 m with occasional tall buildings higher than the measurement height. It was demonstrated that velocity deficits due to tall building wakes were detectable in Velocity Azimuth Display scans at an elevation of 0°. An ensemble averaging approach was used to obtain mean wake characteristics as a function of wind direction and atmospheric conditions.

Two buildings in close proximity to the lidar were studied; one of 90 m height with a slender, symmetrical plan shape, and one of 40 m agl height with a rectangular plan shape. Typical observed wake lengths were 120-300 m and widths were 80-150 m under different atmospheric conditions. Under neutral conditions, wake dimensions were compared with predictions using the ADMS-Build dispersion model. Compared to the observations, the model under-predicted the wake width, length



and velocity deficit, which is consistent with the wind tunnel study of Hertwig et al. (2019) for the same site. This suggests that simple dispersion model assumptions of wake shape and mixing should be revisited for high aspect ratio buildings embedded within urban canopies.

In the field, unstable conditions co-exist with a deeper boundary layer, and stable conditions with a shallower boundary layer, therefore the effect of these influences cannot be separated. Eddy covariance measurements at 191 m on top of a telecommunications tower (BT Tower) combined with the mixing height derived from vertical Doppler lidar scans, were used to define three stability classes. Unstable conditions with a deeper boundary layer produced a shorter wake unless turbulence upstream of a building was enhanced (e.g., by the wake of an upstream building). This is similar to the findings of Hertwig et al. (2019)

under neutral conditions and the wind turbine studies of Aitken et al. (2014) under atmospheric conditions.

The weakened stability dependence was confirmed as the wind direction incident on the buildings changed and there were fewer or more obstacles upstream. This result is understandable as under unstable conditions, stronger mixing reduces the velocity deficit. However, a local enhancement of mechanically produced turbulence due to an upstream obstacle dominates over buoyant production, causing locally neutral conditions in the wake. Stable and shallow boundary layer conditions were

insufficiently sampled (being less common in urban areas) or produced wake dimensions not significantly different to neutral conditions.

Building wakes for the predominant wind direction (south-westerly) showed little sensitivity to wind speed but for most other wind directions the wake length and width increased with increasing wind speed. This is consistent with the previous result that under near neutral conditions (higher wind speeds), turbulence intensity is reduced and wakes are longer than under

unstable conditions.

Site-specific flow patterns due to other roughness elements influenced wake structure. In particular, flow speed-up effects were observed around the wake when wake interference or channelling flow between buildings was apparent. Methodologically, there are several limitations of this study: results may be sensitive to the spatially-averaged radial velocity defined as reference wind speed, although uncertainty ranges have been presented. Resolution depends on viewing angle, which in turn depends on

obtaining permission to locate the instrument on a certain building. Relative height of lidar and building in this study limits the conclusions to one horizontal "slice" through the 3D wake. The ensemble averaging approach should be explored for its relationship to e.g., wind tunnel ensembles with stationary flow.

Future observational work should explore different lidar scan patterns and building-lidar configurations. This raises the question of which archetype is relevant to study: an isolated tall building with a fully developed wake is probably rare within

cities; clusters of similar tall buildings can be found; more common is a sparse array of taller buildings of different shapes where wakes overlap and a deep roughness sublayer is found. For certain cities, e.g., New York, a compact array of tall buildings forms a dense canopy.

The implications for modelling or theoretical work is that for isolated tall buildings, stability effects seem to be more important compared to when simulating clusters of tall buildings, when wake interference appears to dampen stability effects.

In all cases, the presence of an urban canopy at low levels modifies tall building wakes Hertwig et al. (2019). As previous studies (Heist et al., 2009; Brixey et al., 2009) have established that tall buildings also alter nearby flow within the urban canopy, two-



way coupling between the two under different stability conditions and building heating patterns should be a priority in future work.

*Code availability.* Codes will be available upon request

*Data availability.* Data will be available upon request

*Competing interests.* The authors declare no competing interests

*Acknowledgements.* The authors acknowledge the kind permission of London Southbank University (LSBU) and BT for installing equipment. In particular we thank Dr Elsa Aristodemou, Chris Barnes, and Peter Robinson at LSBU. Technical support was provided by Dr Hannah Gough, Jessica Brown, Ian Read, Eric Mathieu, and Maria Broadbridge at University of Reading. Special thanks to Dr William Morrison for
supplying logging software and data for Figure 1. N. Theeuwes was funded under the Engineering and Physical Sciences Research Council (EPSRC) Grand Challenge grant "Managing Air for Green Inner Cities (MAGIC)" [grant number EP/N010221/1] and NWO Rubicon grant "Clouds above the city" [019.161LW.026]. J. Barlow was supported under the EPSRC grant "Fluid dynamics of Urban Tall-building clUsters for Resilient built Environments (FUTURE)" [EP/V010166/1] and equipment was bought under the EPSRC grant "Advanced Climate Technology Urban Laboratory" [EP/G029938/1].



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

Methods of Observation (TECO-98), Instruments and Observing Methods Rep.70, WMO/TD NO. 877, 1998.

Ortiz-Amezcua, P., Martínez-Herrera, A., Manninen, A. J., Pentikäinen, P. P., O'Connor, E. J., Guerrero-Rascado, J. L., and Alados-Arboledas, L.: Wind and Turbulence Statistics in the Urban Boundary Layer over a Mountain–Valley System in Granada, Spain, Remote Sensing, 14, 2321, https://doi.org/10.3390/rs14102321, 2022.

Päschke, E., Leinweber, R., and Lehmann, V.: An assessment of the performance of a 1.5 m Doppler lidar for operational vertical wind
profiling based on a 1-year trial, Atmospheric Measurement Techniques, 8, 2251–2266, https://doi.org/10.5194/amt-8-2251-2015, 2015.

Pearson, G., Davies, F., and Collier, C.: An Analysis of the Performance of the UFAM Pulsed Doppler Lidar for Observing the Boundary Layer, Journal of Atmospheric and Oceanic Technology, 26, 240–250, https://doi.org/10.1175/2008jtecha1128.1, 2009.

Robins, A. and McHugh, C.: Development and evaluation of the ADMS building effects module, International Journal of Environment and Pollution, 16, 161–174, 2001.

Rotach, M. W.: On the influence of the urban roughness sublayer on turbulence and dispersion, Atmospheric Environment, 33, 4001–4008, 1999.

Rye, B. and Hardesty, R.: Discrete spectral peak estimation in incoherent backscatter heterodyne lidar. I. Spectral accumulation and the Cramer-Rao lower bound, IEEE Transactions on Geoscience and Remote Sensing, 31, 16–27, https://doi.org/10.1109/36.210440, 1993.

Sheng, R., Perret, L., Calmet, I., Demouge, F., and Guilhot, J.: Wind tunnel study of wind effects on a high-rise building at a scale of 1: 300,
Journal of Wind Engineering and Industrial Aerodynamics, 174, 391–403, 2018.

Song, J., Fan, S., Lin, W., Mottet, L., Woodward, H., Davies Wykes, M., Arcucci, R., Xiao, D., Debay, J.-E., ApSimon, H., et al.: Natural ventilation in cities: the implications of fluid mechanics, Building Research & Information, 46, 809–828, 2018.

Stewart, I. D.: A systematic review and scientific critique of methodology in modern urban heat island literature, International Journal of Climatology, 31, 200–217, 2011.



Sun, H., Gao, X., and Yang, H.: A review of full-scale wind-field measurements of the wind-turbine wake effect and a measurement of the wake-interaction effect, Renewable and Sustainable Energy Reviews, 132, 110 042, https://doi.org/10.1016/j.rser.2020.110042, 2020.

Theeuwes, N. E., Barlow, J. F., Teuling, A. J., Grimmond, C. S. B., and Kotthaus, S.: Persistent cloud cover over mega-cities linked to surface heat release, npj Climate and Atmospheric Science, 2, 1–6, 2019a.

Theeuwes, N. E., Ronda, R. J., Harman, I. N., Christen, A., and Grimmond, C. S. B.: Parametrizing Horizontally-Averaged Wind and
Temperature Profiles in the Urban Roughness Sublayer, Boundary-Layer Meteorology, 173, 321–348, 2019b.

Trujillo, J.-J., Bingöl, F., Larsen, G. C., Mann, J., and Kühn, M.: Light detection and ranging measurements of wake dynamics. Part II: two-dimensional scanning, Wind Energy, 14, 61–75, https://doi.org/10.1002/we.402, 2011.

Vakkari, V., O'Connor, E., Nisantzi, A., Mamouri, R., and Hadjimitsis, D.: Low-level mixing height detection in coastal locations with a scanning Doppler lidar, Atmospheric Measurement Techniques, 8, 1875–1885, 2015.

Vasiljević, N., Palma, J. M. L. M., Angelou, N., Matos, J. C., Menke, R., Lea, G., Mann, J., Courtney, M., Ribeiro, L. F., and Gomes, V. M. M. G. C.: Perdigão 2015: methodology for atmospheric multi-Doppler lidar experiments, Atmospheric Measurement Techniques, 10, 3463–3483, https://doi.org/10.5194/amt-10-3463-2017, 2017.

Wharton, S. and Lundquist, J. K.: Atmospheric stability affects wind turbine power collection, Environmental Research Letters, 7, 014 005, https://doi.org/10.1088/1748-9326/7/1/014005, 2012.

Wildmann, N., Gerz, T., and Lundquist, J. K.: Long-range Doppler lidar measurements of wind turbine wakes and their inter-action with turbulent atmospheric boundary-layer flow at Perdigao 2017, Journal of Physics: Conference Series, 1618, 032 034, https://doi.org/10.1088/1742-6596/1618/3/032034, 2020.

Wood, C., Lacser, A., Barlow, J. F., Padhra, A., Belcher, S. E., Nemitz, E., Helfter, C., Famulari, D., and Grimmond, C.: Turbulent flow at 190 m height above London during 2006–2008: a climatology and the applicability of similarity theory, Boundary-layer Meteorology,
137, 77–96, 2010.





(a)

(b)

**Figure 1.** MAGIC Project site showing buildings A and B. (a) Building heights in height above sea level (a.s.l.), buildings above 36.5 m a.s.l. are highlighted in light blue. The circles are shown every 100 m away from the lidar (red dot). (b) Panoramic view taken from roof-top where lidar was installed (south-east on left to north-west on right).





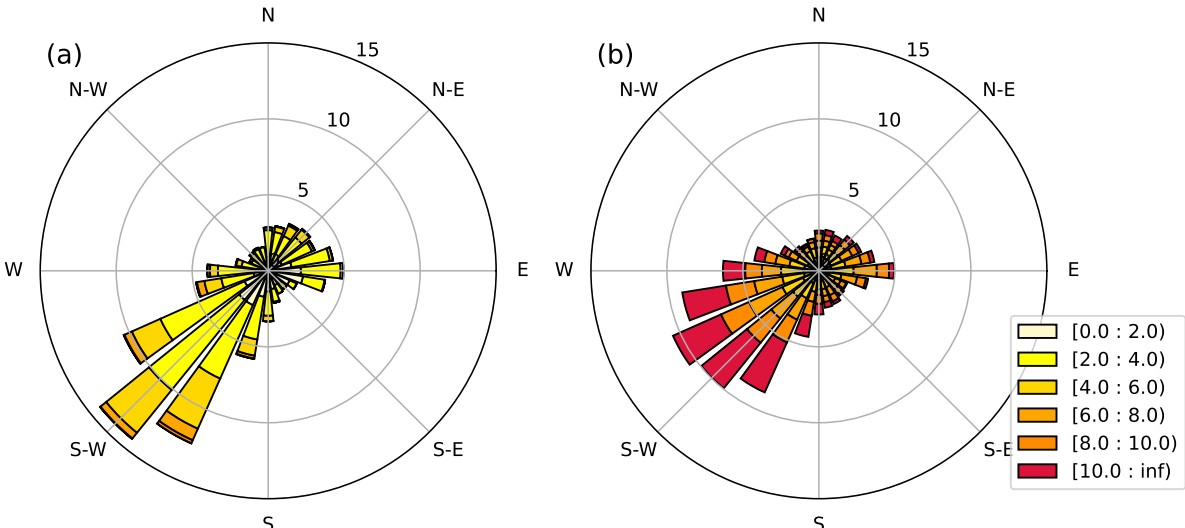

**Figure 2.** Wind rose with frequency of wind direction and wind speed [m s$^{-1}$] derived from Doppler lidar velocities at two heights (a) 36.5 m asl, spatially-averaged wind vectors derived from VAD scans at 0° elevation in a 500 m radius around the lidar and (b) 225.5m asl (222.5 m agl), from VAD scans at 75° elevation. All data between 7 March 2019 and 31 March 2020.

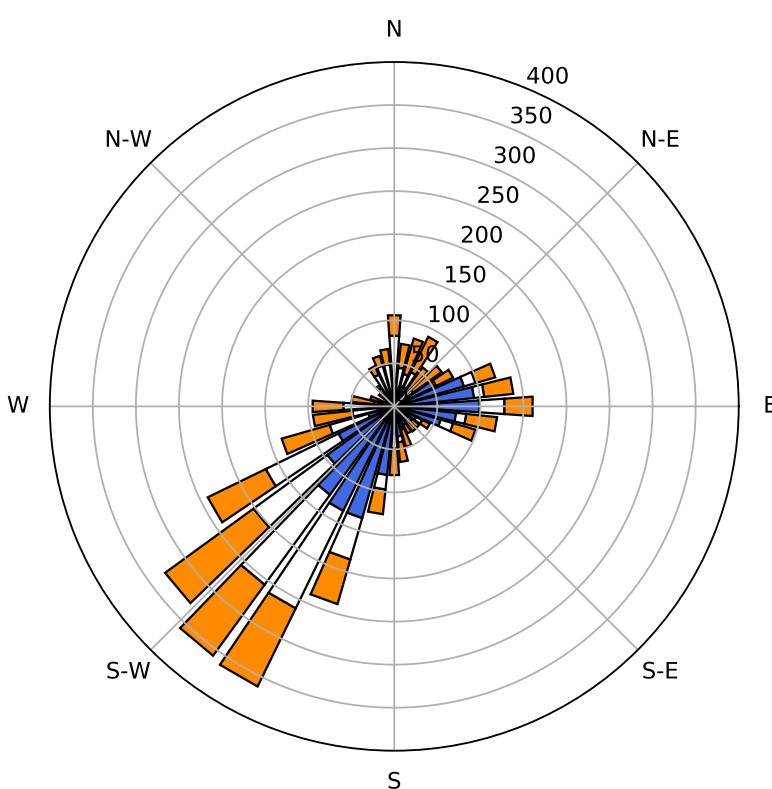

**Figure 3.** Wind rose with boundary layer stability classes for each direction at 36.5 m a.s.l. Boundary layer classes: unstable ($\frac{z-z_d}{L} < -0.1$) and deep ($\frac{H}{z_{\text{MH}}} < 0.1$) (orange); neutral ($-0.1 < \frac{z-z_d}{L} < 0.1$ and moderate $0.1 < \frac{H}{z_{\text{MH}}} < 0.5$) (white); stable (($\frac{z-z_d}{L} > 0.1$) and shallow ($\frac{H}{z_{\text{MH}}} > 0.5$) (blue).



**Figure 4.** The (a) instantaneous radial velocity, $V_\theta$, (b) attenuated backscatter, $\beta$, (c) spatially-averaged radial velocity for a radius of 500 m from the lidar, $\overline{V_\theta}$ (d) normalised velocity deficit, $(V_\theta - \overline{V_\theta})/\overline{V_\theta}$, from a VAD scan at 0° elevation and height 36.5 m asl on 14 July 2019 22:52 UTC. Only data with $SNR + 1 < 1.01$ used.



**Figure 5.** The (a) ensemble mean radial velocity, $\langle V_\theta \rangle$, and (b) ensemble mean normalised velocity deficit, $\langle (V_\theta - \overline{V_\theta})/\overline{V_\theta} \rangle$, for a radius of 650 m from the lidar at 36.5 m asl . Buildings are indicated by gates where $\beta > 0.0002 m^{-1} sr^{-1}$ (grey) and $\beta > 0.001 m^{-1} sr^{-1}$ (black). Data filtered for the predominant wind direction ($210 < \phi < 230$, $n = 1752$), only data with $SNR + 1 > 1.01$ used.

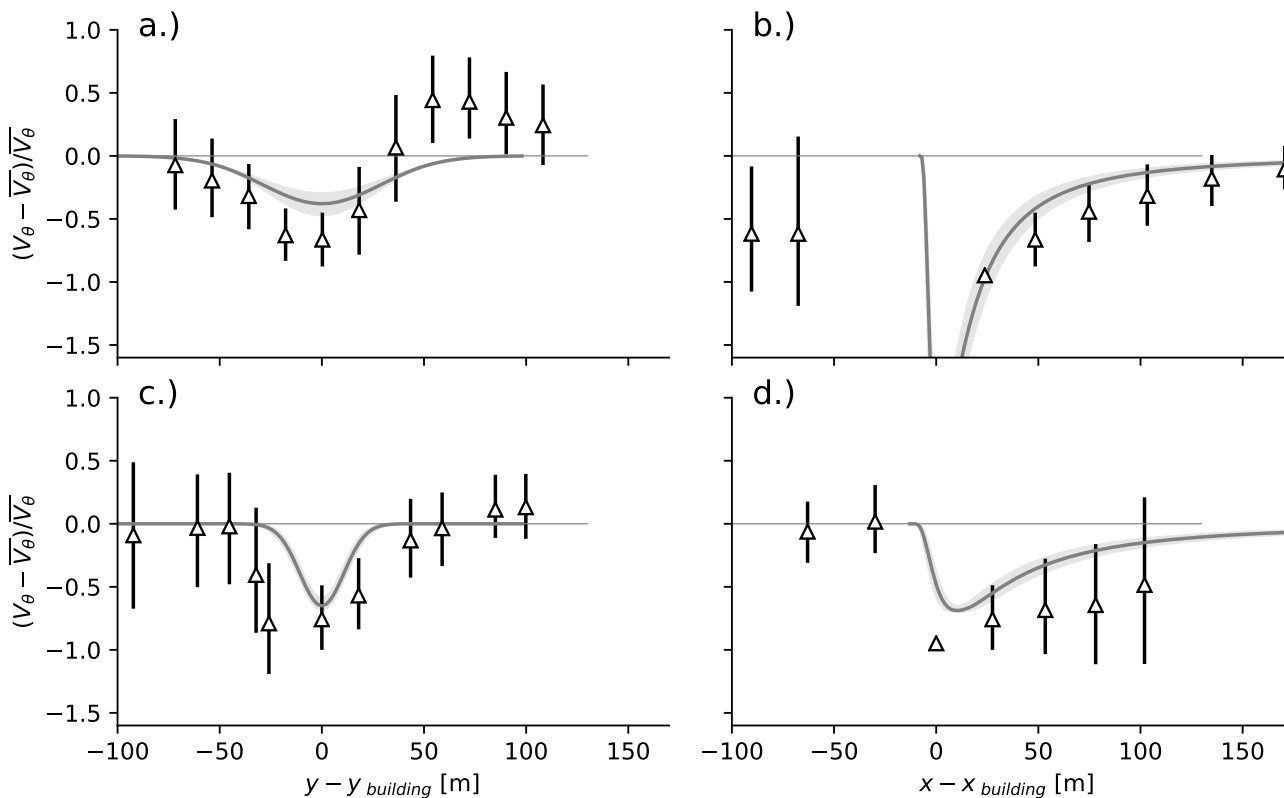

**Figure 6.** Cross-sections through the wakes of building A (a,b) and building B (c,d) perpendicular to the wind direction (a,c) and parallel to the wind direction (b,d) for near-neutral conditions ($-0.1 < \frac{z - z_d}{L} < 0.1$) at BT Tower and LSBU and spatial mean wind velocity $\overline{V_\theta} > 2$ m s$^{-1}$ ($n = 776$), median (triangles) and interquartile ranges (bars) of the measurements and median (lines) and interquartile ranges (shading) of the ADMS-Build predictions. The perpendicular cross sections are (a) 71 m and (b) 32 m away from buildings A and B respectively and are depicted in 8 .

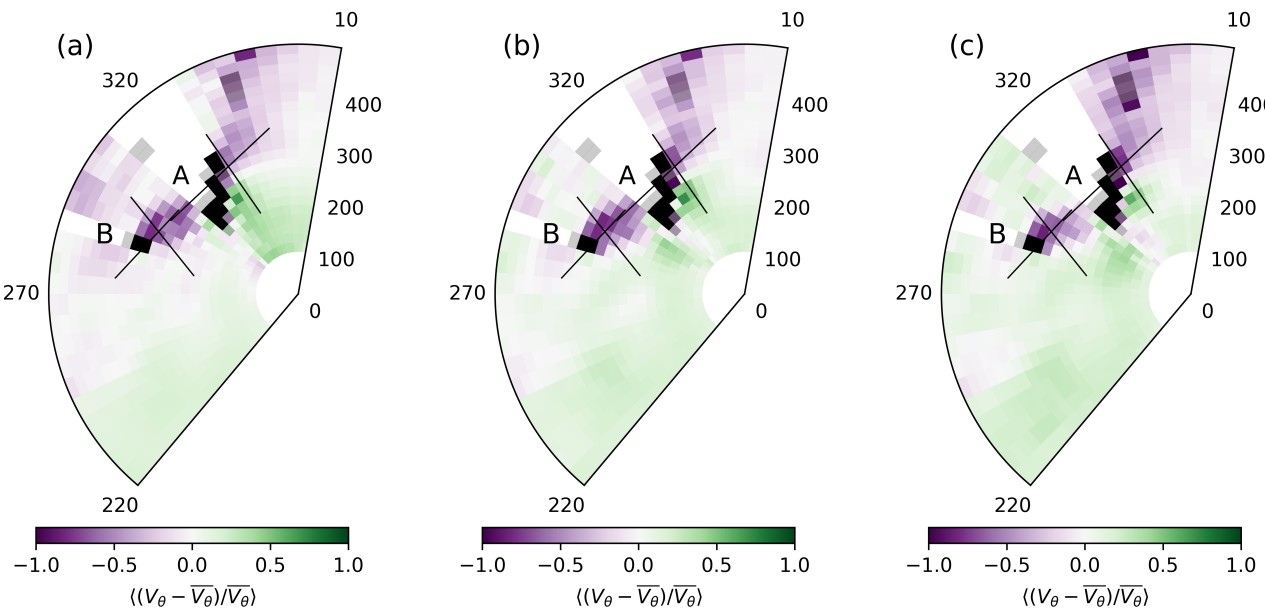

**Figure 7.** Ensemble mean normalised velocity deficit, $\langle (V_\theta - \overline{V_\theta})/\overline{V_\theta} \rangle$, for a radius of 500 m from the lidar at 36.5 asl for three stability classes: (a) stable ($\frac{z-z_d}{L} > 0.1$) and shallow boundary layer ($\frac{H}{z_{\mathrm{MH}}} > 0.5$) ( $n = 246$) (b) neutral ($-0.1 < \frac{z-z_d}{L} < 0.1$ and moderate $0.1 < \frac{H}{z_{\mathrm{MH}}} < 0.5$) class ($n = 255$), and (c) unstable ($\frac{z-z_d}{L} < -0.1$) and deep boundary layer ($\frac{H}{z_{\mathrm{MH}}} < 0.1$) ($n = 226$). The gates highlighted in grey and black are where $\beta > 0.0002$ and $\beta > 0.001$ respectively. Data filtered for the predominant wind direction ($210 < \phi < 230$), only data with $SNR > 1.01$ used.



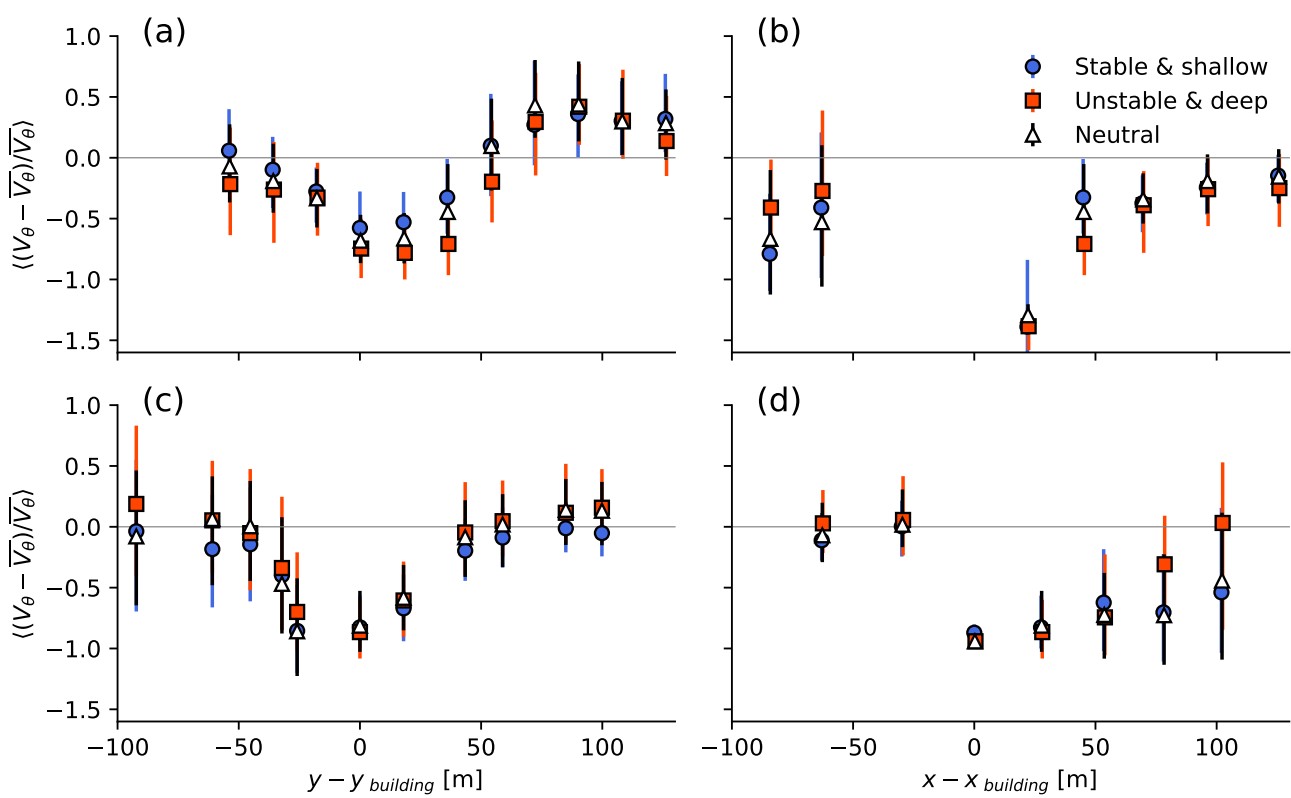

**Figure 8.** Cross-sections through the wakes of building A (a,b) and building B (c,d) perpendicular to the wind direction (a,c) and parallel to the wind direction (b,d) for three stability classes: Unstable ($\frac{z-z_d}{L} < -0.1$) and deep boundary layer ($\frac{H}{z_{\mathrm{MH}}} < 0.1$) (orange squares, $n = 226$), neutral and moderate ($-0.1 < \frac{z-z_d}{L} < 0.1$ and $0.1 < \frac{H}{z_{\mathrm{MH}}} < 0.5$) class (white triangles, $n = 255$), and stable (($\frac{z-z_d}{L} > 0.1$) and shallow boundary layer ($\frac{H}{z_{\mathrm{MH}}} > 0.5$) (blue circles, $n = 246$). The perpendicular cross section are (a) 71 m and (b) 32 m away from buildings A and B respectively.




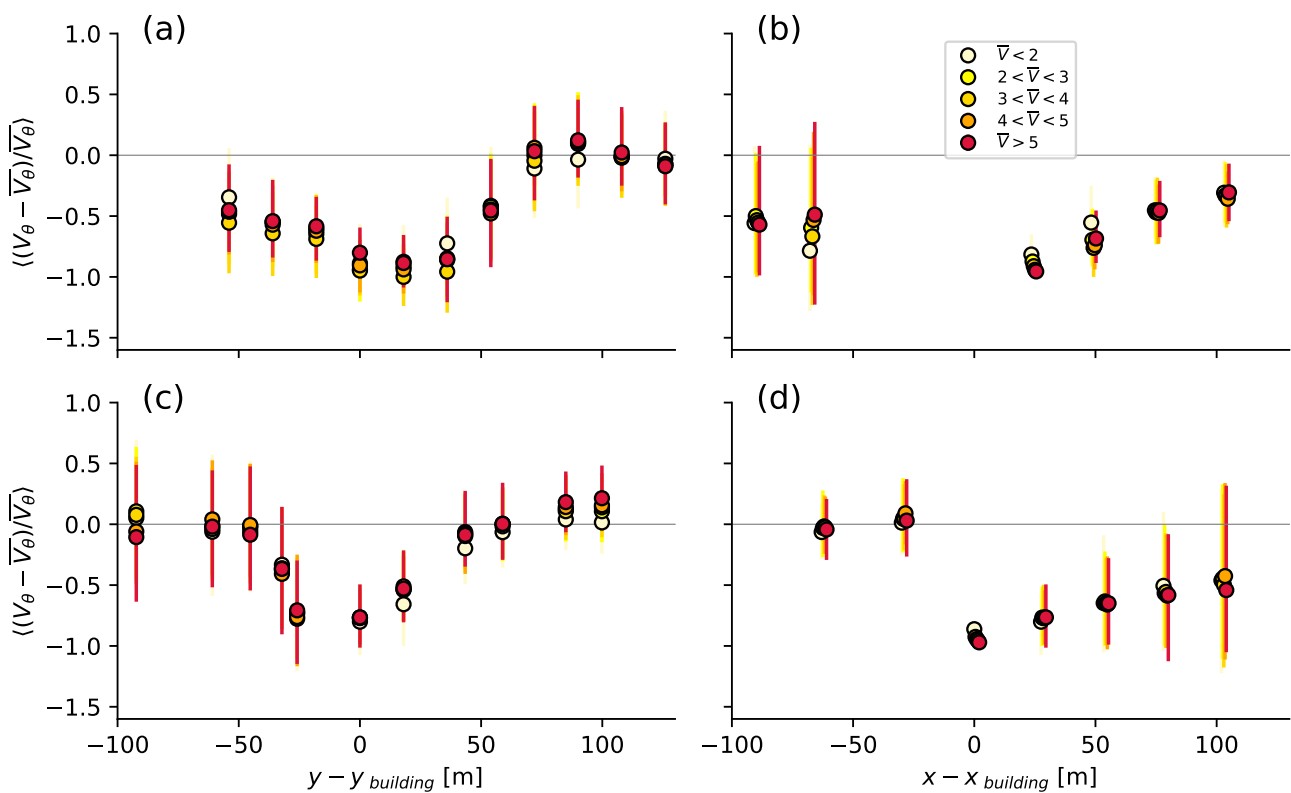

**Figure 9.** Cross-sections through the wakes of building A (a,b) and building B (c,d) perpendicular to the wind direction (a,c) and parallel to the wind direction (b,d) for five wind speed classes (markers coloured from light yellow to red): $\overline{V} < 2$ ($n = 5296$), $2 < \overline{V} < 3$ ($n = 5896$), $3 < \overline{V} < 4$ ($n = 3737$), $4 < \overline{V} < 5$ ($n = 1993$), $\overline{V} > 5\ ms^{-1}$ ($n = 1586$). The perpendicular cross section are (a) 71 m and (b) 32 m away from buildings A and B respectively.




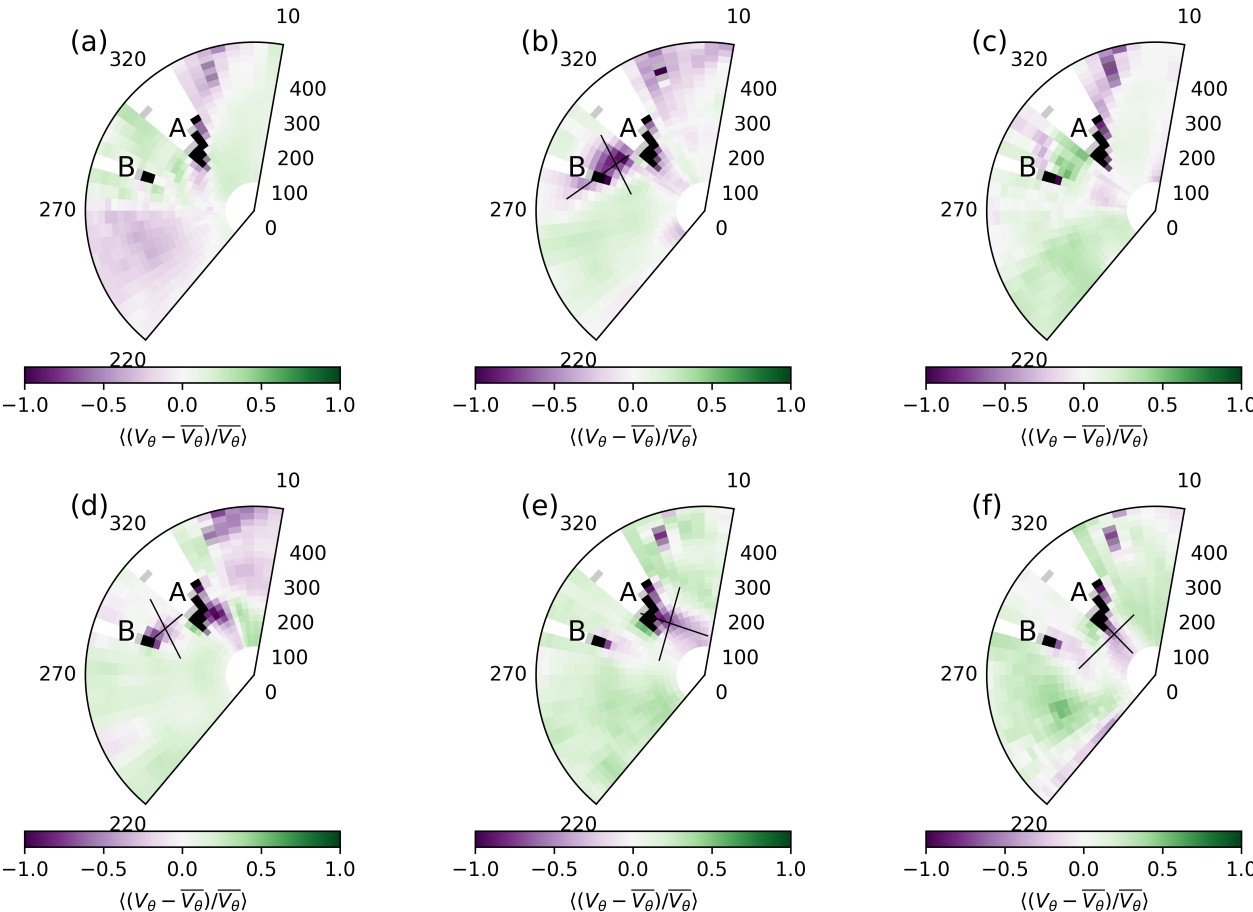

**Figure 10.** Normalised velocity deficit $((V_\theta - \overline{V_\theta})/\overline{V_\theta})$ for a radius of 500 m from the lidar at 36.5 m asl for 6 wind directions: (a) $0°$ ($n = 694$), (b) $60°$ ($n = 838$), (c) $180°$ ($n = 821$), (d) $250°$ ($n = 1486$), (e) $290°$ ($n = 453$), (f) $315°$ ($n = 375$). Data filtered for $10°$ bin centred on each wind direction and $SNR > 1.01$. Grey lines indicate the cross-sections shown in Figures 11 and 12





**Figure 11.** Cross-sections through the wakes perpendicular to the wind direction (a,c,e,g) and parallel to the wind direction (b,d,f,h) for three stability classes: Unstable ($\frac{z-z_d}{L} < -0.1$) and deep boundary layer ($\frac{H}{z_{\mathrm{MH}}} < 0.1$) (orange squares); neutral ($-0.1 < \frac{z-z_d}{L} < 0.1$ and moderate $0.1 < \frac{H}{z_{\mathrm{MH}}} < 0.5$) class (white triangles), and stable (($\frac{z-z_d}{L} > 0.1$) and shallow boundary layer ($\frac{H}{z_{\mathrm{MH}}} > 0.5$) (blue circles) for 4 wind directions: (a,b) building A at $60°$ ($n = 62, 14, 111$); (c,d) building B at $250°$ ($n = 121, 93, 71$); (e,f) building A at $290°$ ($n = 31, 12, -$); and (g,h) building A at $315°$ ($n = 19, 24, 11$). The perpendicular cross-sections are (a) 97 m (c) 62 m (e) 65 m and (g) 72 m away from each building.







**Figure 12.** Cross-sections through the wakes perpendicular to the wind direction (a,c,e,g) and parallel to the wind direction (b,d,f,h) for five wind speed classes: $\overline{V} < 2$; $2 < \overline{V} < 3$; $3 < \overline{V} < 4$; $4 < \overline{V} < 5$; $\overline{V} > 5$ (from light yellow to red) for 4 wind directions: (a,b) building A at $60°$ ($n = 217, 354, 148, 32, 39$); (c,d) building B at $250°$ ($n = 257, 361, 339, 205, 291$); (e,f) building A at $290°$ ($n = 92, 171, 80, 58, 27$); and (g,h) building A at $315°$ ($n = 110, 153, 63, 17, 11$). The perpendicular cross-sections are (a) 97 m (c) 62 m (e) 65 m and (g) 72 m away from each building.