# Peer review of "Observations of Tall-Building Wakes Using a Scanning Doppler Lidar"

_EGUsphere, 2024_

## Author Comment (AC1)

Author's response to Reviewer #1

We thank all referees for their comments which have helped us to clarify the text.

1. *The manuscript in the 3rd paragraph introduces how Dopper lidars have been used in wind energy research studies and in the 4th paragraph they highlight results of research investigations of wind turbine power production and wakes using Doppler lidars. It is not clear how and if these references are relevant to the research objective of this study. Furthermore, there have a been a couple of already published studies that investigate the impact of buildings or bridges to the surrounding wind conditions. I suggest that the authors should consider mentioning such studies in the introduction if they think that it is relevant. For example:*

- *Kiros E.W. Lim, Simon Watkins, Reece Clothier, Raj Ladani, Abdulghani Mohamed, Jennifer L. Palmer, Full-scale flow measurement on a tall building with a continuous-wave Doppler Lidar anemometer, Journal of Wind Engineering and Industrial Aerodynamics, Volume 154, 2016, Pages 69-75, ISSN 0167-6105, https://doi.org/10.1016/j.jweia.2016.04.007.*

- *Xiaoying Liu, Hongwei Zhang, Songhua Wu, Qichao Wang, Zhiqiang He, Jianjun Zhang, Rongzhong Li, Shouxin Liu, Xi Zhang, Effects of buildings on wind shear at the airport: Field measurement by coherent Doppler lidar, Journal of Wind Engineering and Industrial Aerodynamics, Volume 230, 2022, 105194, ISSN 0167-6105, https://doi.org/10.1016/j.jweia.2022.105194.*

- *Mohammad Nafisifard, Jasna B. Jakobsen, Jonas T. Snæbjörnsson, Mikael Sjöholm, Jakob Mann, Lidar measurements of wake around a bridge deck, Journal of Wind Engineering and Industrial Aerodynamics, Volume 240, 2023, 105491, ISSN 0167-6105, https://doi.org/10.1016/j.jweia.2023.105491.*

*For the record, I am neither the author nor any of the co-authors of the above publications.*

We thank the reviewer for these articles, we were not aware of them. We have added the last two references to the manuscript and added a sentence at the start of the 5th paragraph to make it clearer why we refer to wind turbine wake studies, which are extensive.

1. *To estimate the reference wind speed with the current measuring setup it was assumed that the mean wind speed and direction is the same over a circular area with a radius of 500 m. I understand that this is a necessary assumption, but probably it is not realistic. This is partially discussed in the lines 336 – 342. I think that this should be clearly stated as one the limitations of the used measuring setup.*

We agree that it is not realistic to assume that mean wind speed is the same over a 500m radius area, but we respectfully point out that we do not do this: instead, we compute a spatially-averaged wind velocity, which is a standard approach in studies of roughness sub-layer winds. We've added the following sentence in section 4 (analysis of the Horizontal VAD scans): "We acknowledge this makes the comparison of the magnitude of the velocity deficit with previous studies difficult. However, measuring in an urban area limits the possibilities for measuring undisturbed flow."

2. *The scanning Doppler lidar measures the projection of the wind vector on each line-of-sight of the scanning pattern. The wind direction of the case study presented in Figs. 4, 5 and 6 was 220 degrees. For that wind direction the line-of-sight directions is almost perpendicular to the mean of the wake of the buildings A and B. This measuring configurations makes the estimation of*

*the radial speed uncertain since small errors in either the pointing accuracy or in the radial speed estimation can result to large errors in the instantaneous radial speed when normalizing with a mean Vtheta that has almost a zero value. I think that this point should be highlighted in either Section 5 or in a "Discussion" section.*

We agree that this is a significant uncertainty for south-westerly direction winds studying wakes from these particular buildings (which was the scientific objective of the MAGIC project). Our approach of taking an ensemble mean velocity deficit may offset some of this uncertainty if it is statistically random: in the mean over many samples, the wake appears as a coherent flow feature, and we use the ensemble spread to define an uncertainty (e.g., see Figure 11). The "ideal" wind directions (e.g. a north-westerly, see Fig 11) are very rare, introducing error to the ensemble mean velocity deficit due to a limited ensemble. Certainly, compared to a wind turbine study, where the lidar can be placed at an optimal location with respect to the orientation of the wake, errors are higher - however, in urban studies, instrument location is often a compromise. We have added some text to paragraph 5 of section 4 to acknowledge the error:

"It is acknowledged that there might be large errors in instantaneous normalised velocity deficits for this wind angle as the line of sight radial velocity may be near-zero due to the lidar beam scanning perpendicular to the wind direction in the building wakes.  However, in taking an ensemble mean, a coherent wake structure emerges."

and text in the conclusions section to highlight the need to better optimise instrument-wake location in future studies:

"Errors in velocity deficits could be reduced by selecting scanning directions aligned with the prevailing winds, if an optimal building-lidar configuration can be found"

*3.          It is not very clear how are the data of the ADMS-Build model used to compare to the measurements. Is the wind vector estimated from the ADMS-Build model projected to each line-of-sight? Section 5 is written now in a way that it could be understood that the mean radial speed is the input to the ADMS-Build model.*

This is a mistake, we apologise - the symbol has been corrected to be |V|, not $\bar{V}$.

*4.          I suggest adding in the caption of the figures that present cross-sections of the wake flow, that these profiles correspond to the normalised radial velocities in the wake.*

We acknowledge that the caption text could be clearer, and have modified this for Figures 8, 9, 11 and 12.

*5.          I think that the "Conclusions" section is quite lengthy. I suggest considering if it would make more sense splitting this section to a "Discussion" and a "Conclusions" section.*

We've moved two of the paragraphs in the conclusion section to a separate section "Implications for further field studies". This reduces the conclusions and adds a section dedicated to recommendations for further observational campaigns.

*Specific Comments*

1. *Line 73. What kind of data were used to estimate the mean canopy height and for plotting Fig. 1? Furthermore, please explain how the plan area index is calculated.*

The digital elevation model generated from google street view, and these were processed using UMEP. We've added text describing this in the first paragraph or the site description.

2. *Line 86. I think that the authors mean that the "focus distance" was set to infinity not the "focal length".*

Indeed this is the case. This has been changed in the manuscript.

3. *Line 87. The authors write: "The lidar was configured with a range resolution of 18 m with 6 points per range bin". I guess that a range gate was estimated for every 18 m, but what was the length of each range gate? The length of the range gate determines the spatial resolution of a Doppler lidar. Also, what is meant when it is written that each range bin had 6 points?*

We have clarified this sentence:

"The lidar was configured with a non-overlapping range gate length of 18 m."

4. *Lines 95 – 96. The "integration time per ray" mean the averaging time per line-of-sight in the staring mode?*

Yes this partly correct - when the lidar beam pauses at a certain angle to make a measurement (in any mode - vertical staring, or VADs - this is a "ray"), it "stops and stares" for the integration time. We have clarified the sentence:

"Before 9 September 2019 the integration (or averaging) time was 1 s…"

5. *Line 97. It is written "The maximum range was changed to 555 gates…". Do this mean that the maximum range was 555 x 18 m = 9990 m? Please clarify.*

Yes this is the case. We have added this to the text.

6. *Lines 100 – 102. I suggest moving this paragraph at Line 95. As it is written in the current version the duration of each scanning mode is described before they are introduced. Also, mention here that the total duration of the scanning is 30 minutes. Finally, when the authors write 6-point VAD and 72-point VAD do they refer to the azimuth angles or the range gates?*

We have moved this paragraph up and changed 6-point and 72-point to 6-beam and 72-beam, for clarity

7. *Line 166. The lowest gate is at 98.5 m or at 108.5 m (36.5 + 4x18)?*

This should be 96.5 agl, (3*18+9 (gate 4 midpoint) + 33.5), this is now corrected in the manuscript

8. *Line 191. The line-of-sight radial velocities measured by the scanning wind lidar shouldn't be equal to Vtheta = Vmean Cos(theta-phi) ?*

This equation is indeed wrong. In fact it should be:

Vtheta = Vmean sin(theta-phi + 3/2 pi)

Now, this is corrected in the manuscript.

*9.		Line 192. I that that here it should be written that the mean V should be projected to each line-of-sight, not the mean Vtheta.*

We agree that this sentence was confusing and have slightly rephrased it:

"Projecting across all gates gives the spatially-averaged radial velocity field V pertaining to the spatially-averaged velocity V…"

*10.		Line 197. How is the backscattering coefficient calculated in the Doppler lidar measurements? Also, please write the units with roman fonts.*

The backscatter has been calculated using Manninen et al, 2016. This (including the units) is now updated in the manuscript.

*11.		Lines 197 – 199. Are there trees in the measuring area that are high enough to block the Doppler lidar measurements?*

Fortunately the trees in this area are not high enough. Trees can also be seen in figure 1.

*12.		Line 205. How many VAD scans were averaged to plot Fig. 5?*

These are 1752 scans, as indicated by the n=1752 in the figure caption.

*13.		Lines 213 – 217. I suggest drawing the location of the building presented in this paragraph on Fig. 4d, so it can be become more straightforward to relate the features presented in the figure to the building discussed in the paragraph.*

The two buildings that we discuss in these lines discussing Figure 5 are at just over 500 and 580 m away from the lidar - Figure 4 radius is only out to 500 m, unfortunately it is not possible to indicate them on Figure 4d.

*14.		Lines 235 – 244. In the ADMS-Build model is it considered that the inflow of the building A is affected by the wake of the building B?*

No, since we used the spatial mean wind speed in a radius of 500 m around the lidar the wakes are implicitly included in the spatial mean. However we do not explicitly include the effect of building B on building A, since ADMS is only designed for individual wakes. It is difficult to include the wake of the upstream building since the flow is inhomogeneous, the reduction in the wind speed wake varies down to 0.5 m s-1 (figure 6d). Sensitivity tests using ADMS showed that including a reduction of 0.5 m s-1 in the boundary layer wind speed led to a reduction of about 0.1 m s-1 in the velocity deficit.

*15.		Lines 259 – 260. The observed asymmetry of the wake profile couldn't be related to an inhomogeneous inflow due to the wake of the building B?*

This is a possibility, however with the observations we have we cannot prove this. In addition there is also a large street (north–south oriented) that could cause channeling, enhancing the flow on the southeast side of building A. We have added a sentence in the results sections explaining these possibilities at the end of section 6.1.

*16.		Lines 273 – 274. What is meant here with the "sufficient" scans. How many scans were averaged?*

In all captions the number of scans is included as (n=xx). For 'sufficient' scans, we have chosen an arbitrary value of 10 scans. If there are 10 or less in a certain stability/wind speed/direction class, these classes are excluded from the analysis. A sentence is added to the first paragraph of section 6.3

*17.     Lines 314 – 315. How is the length and width of the wake defined in the study? If I am not mistaken this the first time that numerical values of these two characteristics of the wake are mentioned. Please elaborate more about these characteristics in the main text of the manuscript.*

We have added a few sentences quantifying the wake characteristics when the first plots are introduced in section 5.

*Minor Corrections*

*The manuscript is very well written, so I have only three suggestions for minor corrections.*

*1.     Would it be possible to also add the information of the spatial dimension of Fig.1 (top) in meters? For example, by adding this information at the top x-axis and right y-axis. I think that it would help a lot in understanding the distance of the buildings in relation to the location of the wind lidar.*

The circles in the top figure in figure 1 give the distance from the lidar in metres. Each circle is 100 m. We would prefer to keep the lat/lon coordinates on the x and y axis for readers unfamiliar with the measurement site. We have reworded the caption of figure 1 to make this clearer.

*2.     Caption of Figure 6. Is it correct that are "depicted in 8"?*

No this is indeed not correct. This should be figure 7. Now corrected in the caption.

*3.     Please write all units with roman fonts.*

This has now been corrected in (hopefully) all places.

---

## Author Comment (AC2)

Author's response to Reviewer #2

We thank all referees for their comments which have helped us to clarify the text.

*It is a thorough investigation dealing with the ability of DWL to detect wakes from high-rise buildings in an urban environment. I have just a few comments that I ask the authors to consider. Considering the manuscript in the introduction is partly focused on atmospheric dispersion in an urban area with high-rise buildings, I suggest adding a reference to "the Salt Lake City URBAN 2000 Tracer Experiment" which deals with the problems from a dispersion point of view of urban area with several high-rise buildings. A reference could be Allwine, K. J., J. H. Shinn, G. E. Streit, K. L.Clawson, and M. J. Brown, 2002: Overview of URBAN 2000: A Multi-Scale Field Study ofDispersion Through an Urban Environment, Bull.Amer. Meteor. Soc. 83(4), 521-536. Or Hanna Steven, Britter RFEX and Franzese (2003) Baseline urban dispersion model evaluated with Salt Lake City and Los Angeles data.Atmospheric Environment 37(36):5069-5082 DOI: 10.1016/j.atmosenv.2003.08.014*

We have added a reference to the overview paper of URBAN 2000, since here also doppler lidar measurements are described

*Line 114: Do you have a reference for the 30% lower threshold value for the number of data points available? I find the value rather low.*

The sampling frequency was rather high throughout the period (1 Hz and 0.3 Hz later on). However, especially during the 1Hz period we had problems with sufficient SNR at the top of the boundary layer. When using 30% of data availability, we still use 180 or 540 data points to calculate the variance per 30 minute period. For higher order statistics this may not be sufficient. However, we do not use skewness and kurtosis to derive the mixing height.

*Line 117: what is the spread of the 21 values of the mixing height? Did you consider the effect of clouds and rain?*

Half hour periods with rain are filtered out. Clouds are only implicitly excluded, since the signal dramatically reduces inside the clouds. After taking this into consideration, in the convective and in the stable boundary layers the spread in these 21 values is not large and usually around zero (i.e., all 21 threshold values result in the mixing height being at the same range gate at 18 m gate resolution). However, in the transitions (morning and evening) there could be some spread. There is an example of the spread in figure 7 of Barlow et al., 2015. We've added these details in section 2.2 (last paragraph).

*Line 152  How was the threshold values for the stability classes obtained?*

The classes were defined so that each class would contain more than 15% of the available data and the data would be somewhat evenly distributed. This is now added in the text

*Line 162. Same as in l152 but for the stability based on the mixing height.*

This classification was really meant to relate the depth of mixing to the building height and the roughness sublayer and on the other hand the deep convective boundary layer. Twice the building height is generally considered the minimum roughness sublayer depth. We have added some text here and there in this paragraph to make this clearer.

*Line 191 Check Eq. (1).  Units on left hand side is m/s and on the right hand side s m2/s2. Additionally I am not sure that V has been defined.*

We apologise, this was an error introduced in redrafting - the V was extraneous and has been removed.